# Correlation between 146S Antigen Content in Foot-and-Mouth Disease Inactivated Vaccines and Immunogenicity Level and Vaccine Potency Alternative Test Methods

**DOI:** 10.3390/vetsci11040168

**Published:** 2024-04-08

**Authors:** Yongxia Li, Ruai Yang, Fu Yin, Haisheng Zhang, Guoyuan Zhai, Shiqi Sun, Bo Tian, Qiaoying Zeng

**Affiliations:** 1College of Veterinary Medicine, Gansu Agricultural University, Lanzhou 730070, China; 18193158892@163.com; 2China Agricultural Veterinary Biotechnology Co., Ltd., Lanzhou 730046, China; 3Lanzhou Veterinary Research Institute, Chinese Academy of Agricultural Sciences, Lanzhou 730046, China

**Keywords:** foot-and-mouth disease inactivated vaccine, 146S content, antibody titer, qualification rate of antibody titer, PD_50_, IFN-γ

## Abstract

**Simple Summary:**

Foot-and-mouth disease (FMD) inactivated vaccines play a crucial role in curbing the spread of FMDV. The 50% protective dose (PD_50_) is considered the “gold standard” to assess the efficacy of FMD inactivated vaccines. It is a highly important metric to evaluate vaccine quality. However, assessing PD_50_ requires antibody-negative pigs or cattle, and so it is time-consuming and costly and requires high-level biosafety facilities. The 146S component, which represents intact virus particles containing all the neutralizing epitopes of FMDV, can effectively stimulate the production of protective antibodies in animals. We found that antibody titers and IFN-γ secretion levels at specific time points after immunization were positively associated with 146S contents. Additionally, 146S content showed a positive correlation with PD_50_, with greater PD_50_ values recorded for 146S contents ranging from 4.72 to 16.55 µg/dose. The determination of 146S contents could serve as a new method for potency testing, offering an alternative to animal challenge tests. This approach would not only significantly reduce the cost and duration of vaccine production testing but also minimize the need for animal challenge tests, thereby enhancing animal welfare.

**Abstract:**

To investigate the association between 146S antigen contents in FMD inactivated vaccines and levels of antiviral immunity, this study vaccinated 30 kg pigs with three batches of FMD types O and A bivalent inactivated vaccines. Antibody titers and interferon-gamma (IFN-γ) secretion levels were measured on days 7, 14, 21, and 28 after primary immunization and on days 14 and 28 following booster immunization to assess associations between 146S contents and both antibody titers and IFN-γ secretion levels. Furthermore, 30 kg pigs were vaccinated with 46 batches of FMD type O inactivated vaccines and challenged on day 28, after which PD_50_ values were determined to evaluate the association between 146S content and PD_50_. The findings suggested that antibody titers and IFN-γ secretion levels at specific time points after immunization were positively associated with 146S contents. Additionally, 146S content showed a positive correlation with PD_50_, with greater PD_50_ values recorded for 146S contents ranging from 4.72 to 16.55 µg/dose. This investigation established a significant association between the 146S content in FMD inactivated vaccines and induced immune response against FMDV, thereby emphasizing its critical role in vaccine quality control. The determination of 146S content could serve as a new method for potency testing, offering an alternative to animal challenge tests.

## 1. Introduction

Foot-and-mouth disease (FMD) is an acute, febrile, highly contagious, and severe infectious disease that affects cloven-hoofed animals, such as pigs, cattle, and sheep [1,2,3]. It is caused by foot-and-mouth disease virus (FMDV). This disease poses a significant threat to the livestock industry, leading to substantial economic losses and social consequences [4,5,6,7,8]. The World Organization for Animal Health classifies it as a notifiable animal disease [9]. FMD is prevalent in two-thirds of countries worldwide, particularly in Asia, Africa, and the Middle East, and it has gained widespread attention [10,11,12]. Countries free from the disease invest heavily in prevention measures, while affected countries allocate significant resources to its control [13]. Among the seven serotypes of FMDV, types O and A are the most common and damaging [14].

Currently, vaccination with inactivated FMD vaccines is the most effective measure for preventing and controlling this disease [15,16,17]. It plays a crucial role in curbing the spread of FMDV. The 50% protective dose (PD_50_) is considered the “gold standard” for assessing the efficacy of inactivated FMD vaccines [18]. It is a highly important metric used to evaluate vaccine quality. However, assessing PD_50_ requires the use of antibody-negative pigs or cattle, and so it is time-consuming and costly and requires high-level biosafety facilities. The 146S component, which represents intact virus particles containing all the neutralizing epitopes of FMDV, can effectively stimulate the production of protective antibodies in animals [19,20]. Therefore, evaluating the correlation between the 146S content of a vaccine and the PD_50_ is significant.

Research conducted by Black et al. in 1984 laid a strong foundation for subsequent vaccine efficacy testing by establishing a correlation between average antibody titers in vaccinated animal groups and protection against challenge [21]. In 2020, Al Amin et al. further explored the association between antibody titers following vaccination with inactivated FMD vaccines and vaccine efficacy [22]. However, there are no systematic studies examining the correlation between the 146S content in inactivated FMD vaccines, antibody titers, interferon-gamma (IFN-γ) secretion levels, and PD_50_.

This study aimed to evaluate the induced antibody titers and IFN-γ secretion levels after primary and booster vaccinations by administering different doses of bivalent inactivated FMD vaccines (types O and A) with varying 146S contents per dose to pigs. The associations between 146S content and antibody titers, qualification rate, and IFN-γ secretion levels were analyzed. Additionally, the study investigated the association between 146S content and PD_50_ by measuring the PD_50_ values in pigs vaccinated with 46 batches of type O FMD inactivated vaccines with different 146S contents per dose. The findings of this study support the development of a rapid, simple, and cost-effective method to replace the current animal challenge test for assessing vaccine potency. This approach would not only significantly reduce the cost and duration of vaccine production testing but also minimize the need for animal challenge tests, thereby enhancing animal welfare.

## 2. Materials and Methods

### 2.1. Viruses and Vaccines

The FMDV serotype O strain O/BY/CHA/2010 (GenBank JN998085.1) and the serotype A strain A/GDMM/CHA/2013 (GenBank KF450794.1) were provided by China Agricultural Veterinary Biotechnology Co., Ltd. (Lanzhou, China).

The vaccines employed in this study were commercial monovalent and bivalent batches, produced by different manufacturers: 46 batches of the serotype O monovalent vaccine were obtained from China Agricultural Veterinary Biotechnology Co., Ltd. and 3 batches of the serotype O and A bivalent vaccines were obtained from a commercial market. The vaccines were prepared from viruses which had been grown in BHK suspension cell cultures and subsequently inactivated with binary ethyleneimine (BEI). The inactivated antigens were adjuvanted with Montanide ISA-206 [23]. The ratio of serotype O to A antigens in bivalent vaccines is 1:1.

### 2.2. Quantification of 146S

The vaccine samples were demulsified as described previously [24]. Briefly, nine volumes of each sample were mixed with one volume of n-butylalcohol; each sample was then vortexed thoroughly and centrifuged for 5 min at 4 °C, 5000× *g*. The aqueous phase was collected.

The quantity of the 146S component in each batch vaccine was tested with the sucrose density gradient centrifugation method [25]. The aqueous phase collected from the demulsified sample was directly applied to the top of a 15–45% sucrose gradient. After ultracentrifugation at 35,000 r/min for 3 h, the light absorption values of each band were tested at 259 nm with a continuous UV detector, and an absorption peak map was generated. The 146S content in the aqueous phase was obtained by calculating the absorption peak area. The test was conducted three times, and the final 146S value was derived by averaging the results of the three measurements.

### 2.3. Animal Immunization and Challenge

Healthy pigs weighing approximately 30 kg and free of antibodies to FMDV serotypes O and A (LPB-ELISA antibody titers ≤1:8 and seronegative for FMDV non-structural protein 3ABC) were purchased from a designated pig farm in Gansu Province.

For each batch of monovalent vaccine, 17 pigs were randomly divided into 4 groups. Groups 1–3, with 5 pigs in each group, were inoculated intramuscularly, receiving 1 dose (2 mL), 1/3 dose (0.66 mL), and 1/9 dose (0.22 mL) of the vaccine, respectively. Group 4, with 2 animals, was inoculated with PBS and designated as the control group. Twenty-eight days after vaccination, all pigs were inoculated with 1000 ID_50_ of O/BY/CHA/2010 intramuscularly and examined daily for 10 days for clinical signs of FMD. The PD_50_ was calculated based on the Reed–Muench method [26].

For each batch of bivalent vaccine, 34 pigs were randomly divided into 3 groups. Groups 1–3, with 10 pigs in each group, were inoculated intramuscularly with 1 dose (2 mL), 1/3 dose (0.66 mL), and 1/9 dose (0.22 mL) of the vaccine, respectively. Group 4, with 4 animals, was inoculated with PBS and designated as the control group.

Twenty-eight days after vaccination, the animals were boosted with the same vaccine they received in the primary immunization. Blood samples were collected from all animals at 7, 14, 21, and 28 days post-immunization and 14 and 28 days post-booster immunization. Sera were separated for antibody titer assays and IFN-γ level assays.

### 2.4. Serological Assays

#### 2.4.1. Total Anti-FMDV Antibody Assay

The antibody titers of the anti-FMDV A and O serotypes were evaluated following the instructions provided in the kit that was used (FMD type O and A antibody liquid-phase blocking ELISA detection kit, Lanzhou Shouyan Biotechnology Co., Ltd., Lanzhou, China). Briefly, the serum samples were diluted by two-fold dilutions with PBST from 1:8 to 1:1024 and then mixed with an equal volume of FMDV antigen. After incubation overnight at 4 °C, 50 µL of the mixture was transferred to an enzyme-labelled reaction plate and incubated at 37 °C for 1 h. The microplate was washed with PBST, followed by the addition of guinea pig anti-FMDV serum and incubation at 37 °C for 1 h. After washing with PBST, the microplate was incubated with peroxidase-conjugated guinea pig secondary antibodies for 30 min. The test was finally developed with chromogen (TMB) substrate and stopped with 2M H_2_SO_4_. After reading the absorbance at the OD_450 nm_ wavelength using a microplate reader (Bio-Rad, Hercules, CA, USA), the titers were determined by the OD_450_ value 50% of the antigen control.

#### 2.4.2. Interferon (IFN)-γ Assay

The serum IFN-γ secretion levels were evaluated according to the instructions provided in the kit that was used (The pig IFN-γELISA antibody detection kit, Solarbio, Beijing, China). Briefly, standard and serum samples (100 µL/well) were added to 96-well microplates. The microplates were incubated for 60 min at room temperature, followed by the addition of 100 µL working solution of porcine-conjugate anti-Porcine IFN-γ antibody. After incubation for 60 min at room temperature, the plates were washed with buffer, followed by the addition of a working solution of Streptavidin–HRP (100 µL/well). After incubation for 20 min at room temperature, the plates were washed with buffer. The tests were finally developed with chromogen (TMB) substrate and stopped with 2M H_2_SO_4_. After reading the absorbance at the OD_450 nm_ wavelength using a microplate reader (Bio-Rad, USA), the results for the samples were determined by the standard curve generated from the standard samples provided in the kit.

### 2.5. Statistical Analyses

Statistical analysis was performed using two-way analysis of variance, and statistical significance was established at *p* < 0.05 (*), *p* < 0.01 (**), and *p* < 0.001 (***), ns, no significant differences.

## 3. Results

### 3.1. Correlation between the Quantity of 146S and the Antibody Titer after the First Immunization

There were 33.10 µg, 45.30 µg, and 54.00 µg quantities of O + A antigens in the three batches of bivalent vaccines. Within the groups administered one-ninth, one-third, and full doses, the antigen contents (O or A) were 1.84, 5.52, and 16.55 µg; 2.52, 7.55, and 22.65 µg; and 3.00, 9.00, and 27.00 µg, respectively. After immunization, antibody titers were measured on days 7, 14, 21, and 28 using liquid-phase blocking ELISA. The results indicated that for both types O and A, the 27.00 µg and 22.65 µg dose groups showed significantly greater antibody levels than the other groups on day 7 post-immunization (*p* < 0.05). On day 14 post-immunization, the antibody levels in the 27.00 µg and 22.65 µg dose groups were significantly greater than those in the 16.55 µg, 9.00 µg, and 7.55 µg dose groups (*p* < 0.05). The antibody levels in the 16.55 µg, 9.00 µg, and 7.55 µg dose groups were significantly greater than those in the 5.52 µg, 3.00 µg, 2.52 µg, and 1.84 µg dose groups (*p* < 0.05). On day 21 post-immunization, the antibody levels in the 27.00 µg and 22.65 µg dose groups were significantly greater than those in the 16.55 µg, 9.00 µg, 7.55 µg, and 5.52 µg dose groups (O: *p* < 0.01, A: *p* < 0.05). The antibody levels in the 16.55 µg, 9.00 µg, 7.55 µg, and 5.52 µg dose groups were significantly greater than those in the 3.00 µg, 2.52 µg, and 1.84 µg dose groups (*p* < 0.05). By day 28 post-immunization, the antibody levels in the 27.00 µg, 22.65 µg, and 16.55 µg dose groups were significantly higher than those in the 9.00 µg, 7.55 µg, and 5.52 µg dose groups (*p* < 0.05). The antibody levels in the 9.00 µg, 7.55 µg, and 5.52 µg dose groups were significantly higher than those in the 3.00 µg, 2.52 µg, and 1.84 µg dose groups (*p* < 0.05) (Figure 1). Further analysis revealed a moderate correlation [27] between the antibody titers and the 146S contents on day 14 post-immunization (O: R^2^ = 0.6435, *p* < 0.0001; A: R^2^ = 0.6103, *p* < 0.0001) and a strong correlation on days 21 and 28 post-immunization (21 days: O, R^2^ = 0.8417, *p* < 0.0001; A, R^2^ = 0.7672, *p* < 0.0001; 28 days: O, R^2^ = 0.8509, *p* < 0.0001; A, R^2^ = 0.8851, *p* < 0.0001) (Figure 2). In summary, higher 146S doses led to earlier antibody production and higher antibody titers at the same post-immunization times.

### 3.2. Correlation between the Quantity of 146S and the Qualification Rate after the First Imunization

Liu Zhang et al. (2016) reported a strong correlation between liquid-phase blocking ELISA antibody titers and protection against challenge in the context of FMDV vaccination [28]. A protection threshold of 1:64 was established based on complete protection observed in immunized animals with antibody titers at or above this level. The rates of satisfactory O-type antibody titers were then evaluated on days 7, 14, 21, and 28 following the initial vaccination.

On day 7 post-vaccination, the satisfactory antibody titer rates were 20% for the 27.00 µg and 22.65 µg dose groups, while all other dose groups exhibited rates of 0%. By day 14, the rates increased to 80% for the 27.00 µg and 22.65 µg dose groups, while the 16.55 µg, 9.00 µg, and 7.55 µg dose groups had rates of 50%, 30%, and 20%, respectively. The rates for the other dose groups stayed at 0%. On day 21, the percentages in the 27.00 µg and 22.65 µg dose groups reached 100%, while those in the 16.55 µg, 9.00 µg, 7.55 µg, and 5.52 µg dose groups increased to 80%, 70%, 40%, and 20%, respectively. The other dose groups kept at 0%. By day 28, the percentages in the 27.00 µg, 22.65 µg, and 16.55 µg dose groups achieved 100%, and those in the 9.00 µg, 7.55 µg, 5.52 µg, 3.00 µg, and 2.52 µg dose groups rose to 80%, 70%, 50%, 40%, and 30%, respectively. The 1.84 µg dose group still exhibited a 0% increase (Appendix A). The rates of satisfactory A-type antibody titers post-initial vaccination showed a pattern similar to those of the O type (Appendix A). In summary, a higher 146S content resulted in an earlier achievement of satisfactory antibody titers following vaccination, leading to higher rates of satisfactory antibody titers at the corresponding time points.

### 3.3. Correlation between the Quantity of 146S and the Antibody Titer after Boost Immunization

The immunization boost was conducted on day 28 after the initial vaccination, and the antibody titers were assessed on days 14 and 28 after the boost to determine the rates of satisfactory antibody titers. On days 14 and 28 after the boost, the levels of O-type FMDV antibodies induced in the 27.00 µg, 22.65 µg, and 16.55 µg groups were significantly greater than those induced in the 9.00 µg, 7.55 µg, and 5.52 µg groups (*p* < 0.05), while the latter groups showed significantly greater levels than the 3.00 µg, 2.52 µg, and 1.84 µg groups (*p* < 0.05) (Figure 3A). The levels of A-type FMDV antibodies after the boost on days 14 and 28 were similar to those of O-type FMDV antibodies (Figure 3B). Further analysis revealed a strong correlation between antibody titers and 146S contents on days 14 and 28 after the boost (day 14: O, R^2^ = 0.7809, *p* < 0.0001; A, R^2^ = 0.8121, *p* < 0.0001; day 28: O, R^2^ = 0.7677, *p* < 0.0001; A, R^2^ = 0.7609, *p* < 0.0001) (Figure 4). For both the O-type and A-type groups, the rates of satisfactory antibody titers reached 100% on days 14 and 28 after the boost for all dosage groups. In summary, prime–boost immunization elicited a stronger immune response than prime vaccination only, and higher 146S contents resulted in higher antibody titers at the same time points after vaccination.

### 3.4. Correlation between the Quantity of 146S and the IFN-γ Secretion Levels after Immunization

Cell-mediated immunity plays a vital role in the generation of effective immunity and the control of disease following infection and vaccination with FMDV [29,30]. IFN-γ plays a crucial role in this immune response by activating macrophages and natural killer (NK) cells to combat FMDV [31]. Research has shown that measuring the level of IFN-γ released by FMDV-specific T cells has the potential to replace the challenge test in assessing the effectiveness of FMD vaccines [32].

This study aimed to measure the levels of IFN-γ secretion at different time points after primary and booster immunizations by comparing different dosage groups. On day 7 after primary immunization, there was no significant difference in IFN-γ secretion compared to that in all groups (*p* > 0.05). However, on day 14, the 27.00 µg and 22.65 µg dosage groups showing significantly greater levels than the 16.55 µg, 9.00 µg, 7.55 µg, 5.52 µg, 3.00 µg, and 2.52 µg dosage groups (*p* < 0.001), while the latter groups showed significantly greater levels than the 1.84 µg group (*p* < 0.05). By day 21, the 27.00 µg and 22.65 µg dosage groups exhibiting significantly greater IFN-γ secretion levels than the 16.55 µg, 9.00 µg, and 7.55 µg dosage groups (*p* < 0.001). The IFN-γ secretion levels in the 16.55 µg, 9.00 µg, and 7.55 µg dosage groups were significantly greater than those in the 5.52 µg, 3.00 µg, and 2.52 µg dosage groups (*p* < 0.05), and the IFN-γ secretion levels in the 5.52 µg, 3.00 µg, and 2.52 µg dosage groups were significantly greater than those in the 1.84 µg dosage group (*p* < 0.05). On day 28, the 27 µg and 22.65 µg dosage groups showing significantly greater levels than the 16.55 µg, 9.00 µg, and 7.55 µg dosage groups (*p* < 0.05), and the levels in the 16.55 µg, 9.00 µg, and 7.55 µg dosage groups were significantly greater than those in the 5.52 µg, 3.00 µg, 2.52 µg, and 1.84 µg dosage groups (*p* < 0.05). The differences in IFN-γ secretion levels on days 14 and 28 after booster immunization were similar to those observed on day 28 after first immunization (Figure 5).

Further analysis revealed a moderate correlation between IFN-γ secretion and the 146S concentration on day 14 after primary immunization (R^2^ = 0.6995, *p* < 0.0001) and a strong correlation on days 21 and 28 (21 days: R^2^ = 0.8669, *p* < 0.0001; 28 days: R^2^ = 0.7755, *p* < 0.0001) (Figure 6A). A strong correlation was also found between antibody titers and 146S contents on days 14 and 28 after booster immunization (14 days: R^2^ = 0.8505, *p* < 0.0001; 28 days: R^2^ = 0.7598, *p* < 0.0001) (Figure 6B). In summary, higher 146S levels were associated with greater IFN-γ secretion at the same post-immunization time points.

### 3.5. Correlation between the Quantity of 146S and the PD_50_

The vaccines underwent homogenization, followed by separation and purification utilizing sucrose density gradient centrifugation to quantify the 146S content at each vaccine dose. Twenty-eight days after vaccination, the PD_50_ test was performed. The 146S contents in 46 batches of type O foot-and-mouth disease inactivated vaccines ranged from 4.72 to 38.90 µg/dose, with corresponding PD_50_ values ranging from 7.05 to 15.59 (Appendix A). Within the range of 4.72–16.55 µg/dose, higher 146S contents showed a positive correlation with higher PD_50_ values. Furthermore, when the 146S content reached ≥16.55 µg, the PD_50_ values consistently reached a maximum of 15.59. Further analysis revealed a strong correlation between the 146S content of the vaccine and the PD_50_ (R^2^ = 0.8816, *p* < 0.0001) (Figure 7).

## 4. Discussion

FMD has drawn global attention due to its high infectivity rate, high mortality rate among young livestock, and wide host range [33,34]. In response, China has classified it as a Category I animal disease and mandated immunization. Currently, immunization with inactivated vaccines is the most important, cost-effective, and convenient technique for preventing and controlling FMD. Europe successfully reduced and eventually eradicated the disease by implementing this measure [35]. To manage this disease, developing countries continue to prioritize comprehensive prevention and control measures centered on inactivated vaccine immunization [36]. The quality of inactivated vaccines is a critical factor in determining the success of FMD prevention and control [37]. The “gold standard” for testing the quality of these vaccines is the potency test using a target animal challenge. However, this method is time-consuming and costly and involves live viruses, which poses biosafety risks [38]. The effective antigen content, specifically the 146S content in FMD inactivated vaccines, directly determines vaccine quality. The 12S structural subunit is the core component of 146S, produced from 146S particles due to long-term storage (at a low temperature, i.e., 4 °C), acidification, or rupture of cold chains of foot-and-mouth disease inactivated vaccines [39]. The dissociation of 146S into 12S degradation products is accompanied by a decrease in immunogenicity [40] The study by Meloen et al. (1979) showed that the neutralizing antibody reaction triggered by I46S particles was 10 times higher than that of 12S particles [41]. Therefore, establishing the association between the 146S concentration and the anti-FMDV immune response induced by the vaccine will enable a convenient, rapid, and accurate evaluation of the quality and protective efficacy of FMD inactivated vaccines.

Techniques for measuring 146S content include sucrose density gradient centrifugation, high-performance liquid chromatography, sandwich ELISA, complement fixation tests, and cesium chloride density gradient centrifugation. In this study, sucrose density gradient centrifugation, internationally recognized as a technical standard [42] due to its precision, sensitivity, and speed, was employed to determine the 146S contents.

The data demonstrated a positive correlation between the 146S content in FMD inactivated vaccines and antibody titers and IFN-γ secretion levels. A higher 146S content leads to earlier induction of antibody and IFN-γ secretion after initial immunization, as well as higher antibody titers, IFN-γ secretion levels, and rates of satisfactory antibody titers at the same time points post-vaccination. There was a strong correlation between the 146S content and PD_50_. Within the range of 4.72–16.55 µg, a higher 146S content results in higher PD_50_ values. When the 146S content exceeded 16.55 µg, the PD_50_ reached a maximum of 15.59. Thus, evaluating vaccine quality by measuring 146S contents may replace the efficacious testing method of target animal challenge, leading to significant time and cost savings, enhanced animal welfare, and elimination of biosafety risks.

Immunological memory plays a crucial role in booster immunization, where even minimal antigen stimulation can prompt B lymphocytes to differentiate into plasma cells, resulting in a robust immune response and the production of a substantial quantity of antibodies [43]. This study revealed that booster immunization conducted 28 days after primary immunization, even with vaccines with lower 146S contents, could induce a favorable immune response. Following booster immunization with 1.84 µg of 146S on days 14 and 28, all animals achieved qualifying antibody titers, indicating complete protection [28]. These findings have significant implications for clinical immunization strategies.

Research has confirmed that IFN-γ possesses anti-FMDV properties, promoting the activation of NK cells and macrophages and inhibiting FMDV replication within the host [44]. Van Lierop et al. were the first to discover that upon stimulation by FMDV antigens, the body generates IFN-γ via MHC class II-restricted T cells (CD_4_) [45]. Gerner et al. demonstrated lymphocyte production of IFN-γ when cattle infected with FMDV were restimulated with peptides from the FMDV structural protein VP1 region. Yooni Oh et al. also supported the role of both humoral and cellular immunity in the protective efficacy of FMD vaccines [46]. Our findings revealed a positive correlation between the 146S concentration and IFN-γ secretion on days 14, 21, and 28 after primary and booster immunization. A higher 146S content leads to earlier IFN-γ secretion after immunization and greater IFN-γ secretion at corresponding times post-immunization. This discovery may assist in the design and research of new FMD vaccines in the future.

## Figures and Tables

**Figure 1 vetsci-11-00168-f001:**
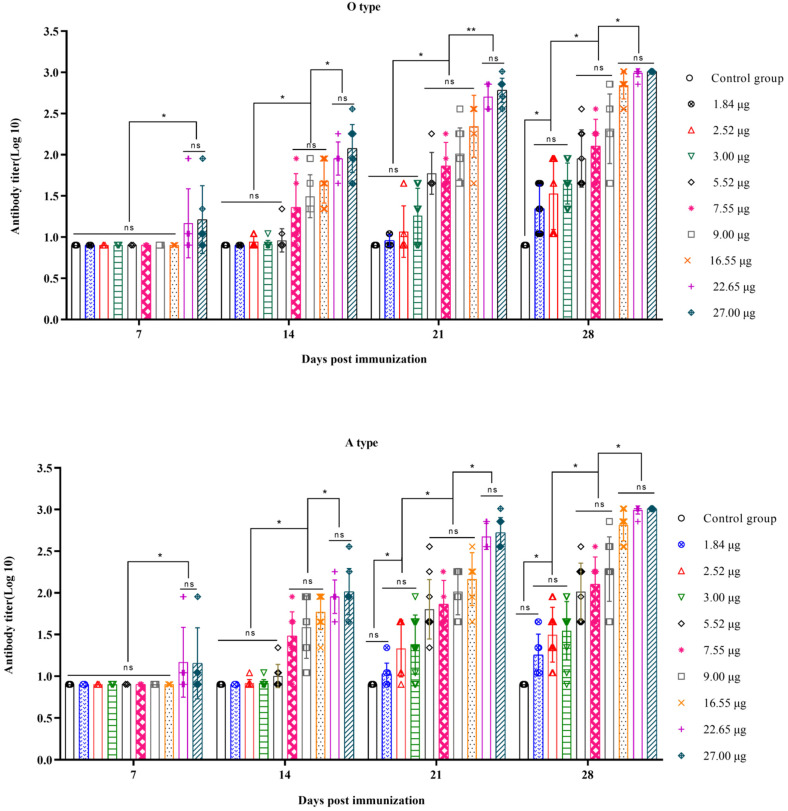
The antibody titers in log base 10 of type O and A following primary immunization at different days post-vaccination with varying 146S doses in the administered vaccines.

**Figure 2 vetsci-11-00168-f002:**
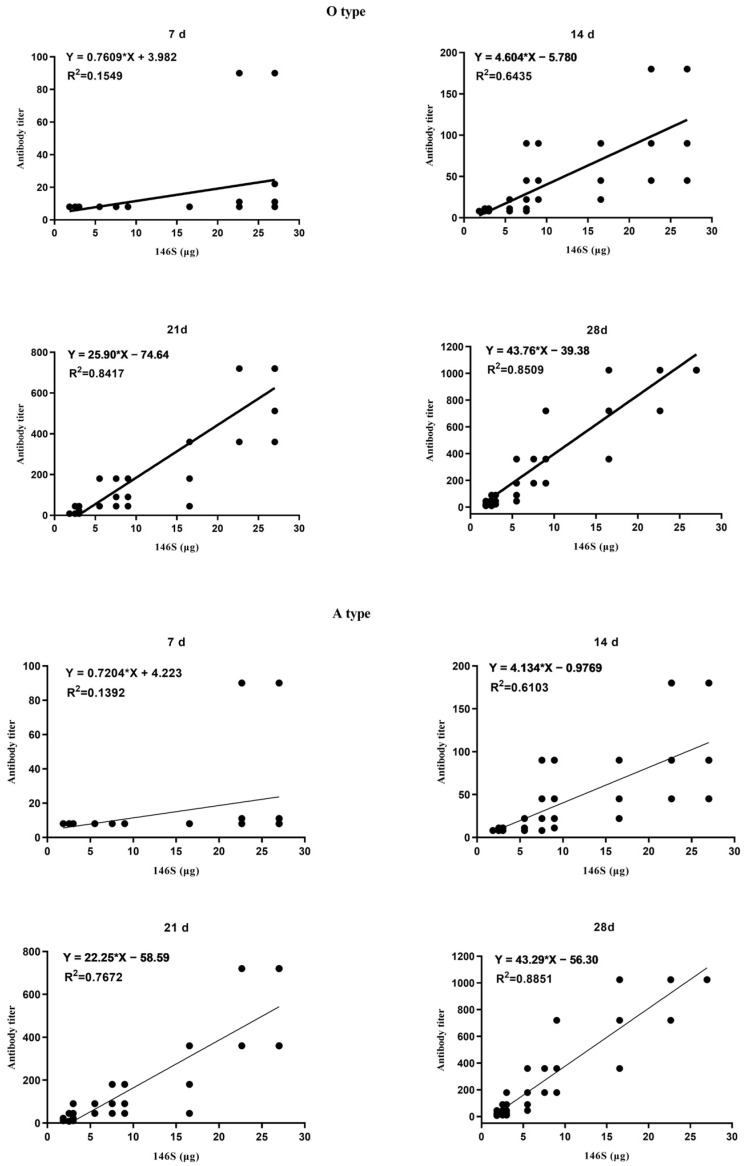
Correlation between 146S dose in the administered vaccine and antibody titers of type O and A at different days post-primary vaccination. The * reresent multiplication sign (×).

**Figure 3 vetsci-11-00168-f003:**
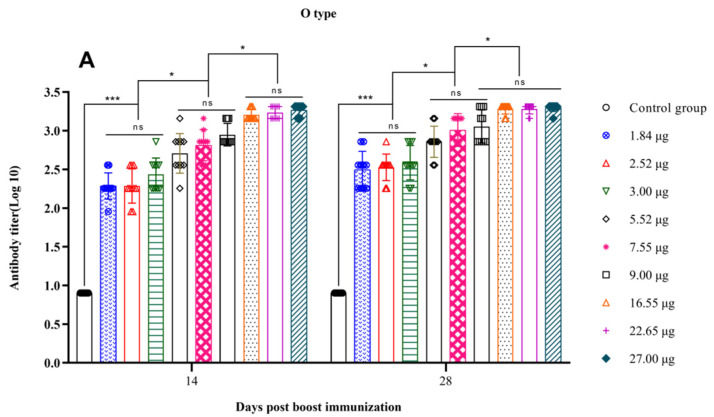
The antibody titers in log base 10 of type O (**A**) and type A (**B**) following booster immunization at different days post-vaccination with varying 146S doses in the administered vaccines.

**Figure 4 vetsci-11-00168-f004:**
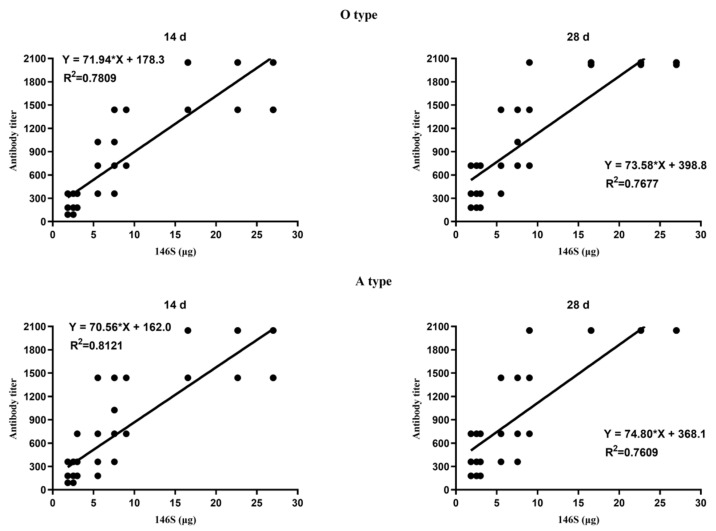
Correlation between 146S doses in the administered vaccines and antibody titers of type O and A at different days post-booster immunization. The * reresent multiplication sign (×).

**Figure 5 vetsci-11-00168-f005:**
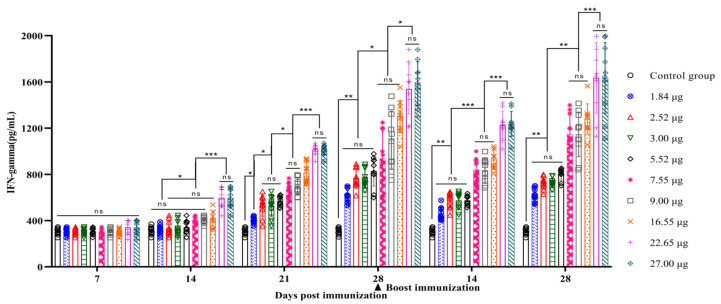
Serum IFN-γ levels following primary and booster immunization at different days post-vaccination with varying 146S doses in the administered vaccines.

**Figure 6 vetsci-11-00168-f006:**
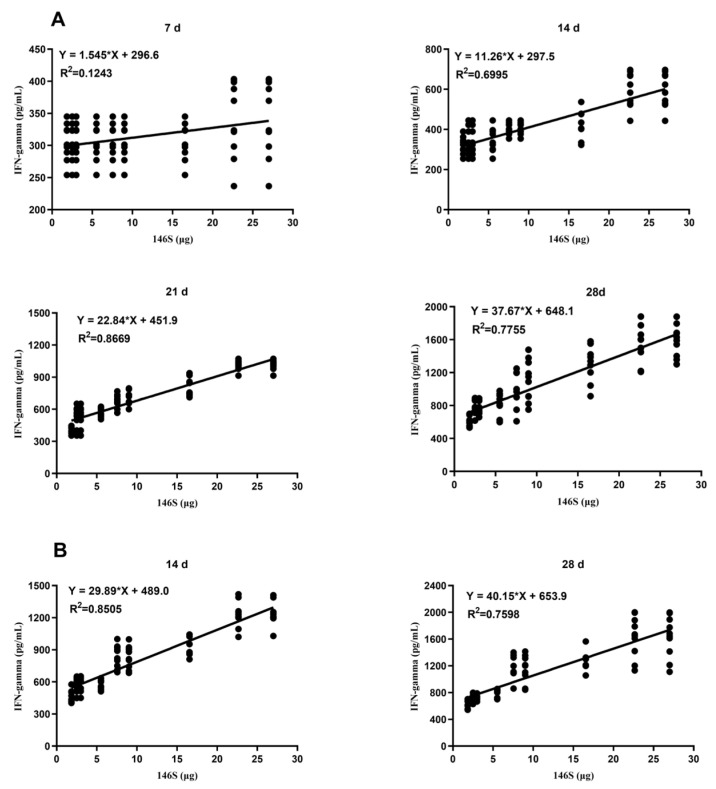
Correlation between the 146S doses in the administered vaccines and IFN-γ secretion at different days after primary (**A**) and booster immunization (**B**). The * reresent multiplication sign (×).

**Figure 7 vetsci-11-00168-f007:**
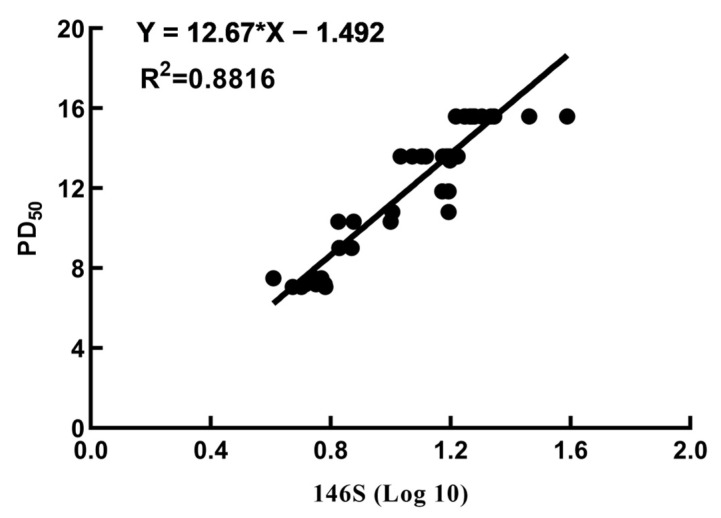
Analysis of the correlation between the log base 10 of 146S doses in the foot-and-mouth disease inactivated vaccines and PD_50._ The * reresent multiplication sign (×).

## Data Availability

All datasets generated for this study are included in the article.

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
