# Peer review of "Correlation between 146S Antigen Content in Foot-and-Mouth Disease Inactivated Vaccines and Immunogenicity Level and Vaccine Potency Alternative Test Methods"

_vetsci, 2024, doi:10.3390/vetsci11040168_

Round 1

Reviewer 1 Report

Comments and Suggestions for Authors

This is an important paper as the quality of FMD vaccines is routinely discussed and alternative methods to looking at the quality need to be developed. 

There are however a few things in the paper that need to be clarified.

Section 3.1 This is a very confusing paragraph to read as it is mostly data. A lot of emphasis seems to be placed on comparing the results to the control group which has received nothing. As the study is looking at more 146S vs less 146S I think the emphasis should be on the differences seen between groups. This would make the paragraph smaller and more easy to read, which capturing the most important data.  The same could be said for section 3.3 and 3.4.

There is also no discussion of 12S and what affect that might have on the immunity. The author should touch on this and highlight what impact 12S might have on vaccination. 

The legends appear to be missing which makes the figures difficult to interpret.  These need to be added.  For instance is the data in log base 10 and I assume they have been carried out by ELISA?

Small changes:

Line 134 - wavelength is spelled incorrectly

Line 139 - briefly is spelled incorrectly 

Line 146 - full stop before After

Line 183 - The reference to figure 2 is not a sentence. 

The font seems to change throughout section 2.4.2 and then becomes much smaller in the result section

Comments on the Quality of English Language

See above for minor changes that need to be made.

Reviewer 2 Report

Comments and Suggestions for Authors

All in all, the paper is interesting and presents quite some data.  It reads well and is structured well. However, some improvements could be made to explain some parts in more detail. Especially in the method section. Please find my minor and major points of attention below.

Minor points

Line 17 & 31: please explain the 4.72-16.55 ug - is it per dose, per ml or per vaccine batch?

Line 109 & 116: please explain what is “1 dose” (how many ml?)

Abstract and introduction refers to piglets, while the method section – to 30 kg pigs (could be considered almost not piglets anymore). Would be better to indicate either the age of the animals or the phase of production to avoid confusion.

Line 155: “doses of 33.10 μg, 45.30 μg, and 54.00 μg, respectively” – more explanation preferred to make the later mentioned antigen doses clearer. The 33.10 ug, 45.30 ug and 54.00 ug in the bivalent vaccines are O+A antigens. So further mentioned dose of 16.55 ug is 33.10/2 (either O or A). 1/9 dose is 1.84 ug.

Line 192: what is the qualification rate? – needs explanation; later in the section authors talk about satisfactory antibody titer that is being compared. Is that the same? Sections 3.2 and 3.3 describe the detected antibody titers after primary and secondary immunization. Why are the titles so different? Additionally, in the discussion part (Line 329) qualifying antibody titers mentioned again.

Line 195-196: difficult to understand what is the 1:64 threshold. Is it from the referenced paper? Or was it established by the authors of this article? How was this ratio established – a more thorough explanation is required. Also, why calculate antibody rate in % for the primary immunization (section 3.2), but for the booster immunization compare titers (section 3.3). Why not do both the same for better comparison?

Line 228-229: part of the sentence “In summary, boost immunization elicited a stronger immune response than initial vaccination,…” is not completely accurate. Of course, that prime-boost approach elicits stronger immune response. That is the point of this approach. Consider re-wording it to: “In summary, prime-boost immunization elicited a stronger immune response than prime vaccination only,… 

For most figures (specifically figures 1, 2, 3, 4, 5, 6 and SF1) replacing “times” with “days post vaccination” and “vaccine concentration” with “content in the administered vaccine” in the provided caption would be an improvement.

Caption Figure 3: serotype A not mentioned in the caption, but panel B shows serotype A data; the caption should reflect the text – Figure 3A and Figure 3B

Caption Figure 6: missing secondary/booster immunization in the caption, but panel B shows data after secondary immunization; the caption should reflect the text – Figure 6A and Figure 6B

Supplementary Fig.1: caption mentions qualification ab rates, see previous comment about this; the caption should reflect the text – Figure S1A and Figure S1B

Major points

Line 11 & 53: sentence “It is an accurate and reliable metric.” regarding the PD50. While it is considered the gold standard in assessing the efficacy of FMD vaccines, there is literature describing low in vivo reproducibility and repeatability: https://doi.org/10.1016/j.vaccine.2006.12.049 . I think, this sentence should be rephrased or at least some of the concerns regarding the method should be mentioned. Also, this paper is cited in the paper (can be found in the reference list), but regarding PD50 being time-consuming, costly, and posing biosafety risks.

Line 64: please include references to some of the “few systematic studies examining the correlation between 146S content in vaccines, antibody titers, IFN-É£, and PD50”

Section 2.1 – requires clarification regarding the antigens and/or vaccines used. It is unclear for what the antigens were provided from The China Agricultural Veterinary Biotechnology Co. Ltd. The next sentence states that the vaccines used for the study were commercially available monovalent and bivalent blends. But further on – that the vaccines were made from inactivated antigens and ISA206 adjuvant. So - were the blends commercially available (bought from different manufacturers) or made by the authors with the provided antigens?

Section 3.5 – paragraph difficult to understand. This is arguably the most interesting part of the paper. Could include in the supplementary file some of the raw data, e.g., what is described here (line 283-284): “Within the range of 4.72-16.55 μg, a higher 146S content showed a positive correlation with higher PD50 values.’ Additionally, why is the 146S content in Figure 7 suddenly given in log, if in the text it is described in ug? Why is the maximum of PD50 15.59 only? Is this the consequence of the calculation method choice (Reed-Muench)? Please explain in the text.

Comments on the Quality of English Language

There are quite some typing mistakes, and sentences that suffer from suboptimal translation that make them difficult to understand. Quite some of these occur in the method section. Below some of the more noteworthy points of attention (simple one-letter typing mistakes not summarized here): 

Line 70: post-vaccination needs to be moved after the “and” (post-vaccination 146S content and post-vaccination antibody titers)

Line 99-100: remove part “the peak area was directly proportional to the concentration of 146S”

Line 100-101: remove part “of the intact FMDV particle”

Line 119: word “vaccine” missing? “.. animals were boosted with the same vaccine as primary immunization.”

Line 119: replace “Four weeks” with “28 days” (as the rest of the paper operates in days post-vaccination)

Line 128-129: part “added equal volume FMDV antigen.” needs rewording

Line 130-131: part “the microplate was added guinea pig anti-FMDV serum” needs rewording

Line134 and 147: can skip word “wavelength”; if not skipping – there is a typo in the word “wavelength”

Line 135-136: part “the titers were determined by the value 50% of antigen control.” Needs rewording and better explanation

Section 2.4.2 – adjust text font/size to be the same as the rest of the paper

Line 139-140: part “Briefly, 96-well plates were added standard and samples” needs rewording

Line 141: part “the microplate was added in 100 ul working solution” needs rewording

Line 148: part “determined by the standard curve” needs more explanation

Line 161, 220, 222, 248, 250, 251, 258: remove word “did”

Line 192: title of section 3.2 should be on the same page as the section itself

Line 336 – check spelling of the referenced authors’ last name

Reviewer 3 Report

Comments and Suggestions for Authors

The authors presented a manuscript describing the correlation between the FMDV 146S content and vaccine potency. 

The comments and suggestions I have are: 

Why are the number of batches of antigens so greatly different (46 batches of monovalent and only 3 batches of bivalent)? 

Why did you only challenge animals with type O? Why not type A as well?

Why did you not standardize the antigen concentrations (eg 1, 2, 3, 5 etc)?   

The figure legend for figure 3 states "antibody titers for type O..." but panel B is labeled A type. 

IFNy production increases in a dose dependent manner, but never seems to reach a plateau. At what concentration would you expect those values to plateau?  

In figure 1, the antibody titers reported from type A and type O are almost exactly the same. Is that accurate? Why do you see such an increase in antibody titers in the control groups at day 28 it both the A and O groups? 

You have made some very sweeping conclusions that this could replace animal challenge models for vaccine testing. While you show there is a correlation here, I don't believe the data are strong enough to make such a conclusion without comparing challenge data from multiple serotypes of vaccinated and challenged animals.  

Comments on the Quality of English Language

Overall, the English was clear and well written. There were spelling/grammatical errors though out the manuscript that should be correct. To point out a few: 

Line 111: erroneous period in the middle of a sentence 

Line 127: "Brefly"

Line 134: "waveleng"

Additionally, the size of the font changes throughout the paper. This should be made consistent. 

There are numerous phrases that are repeated thought out the paper. These should be reviewed for enhance the flow. 

Reviewer 4 Report

Comments and Suggestions for Authors

The authors have conducted a dose-response study by vaccinating pigs with either a full dose, a 1/3 dose, or a 1/9 dose of inactivated FMDV vaccine. Several batches of vaccine were used, each containing a different amount of antigen, giving a wider range of vaccine doses administered from 1.84ug to 27ug. The results show that, following a prime and a boost vaccination, the levels of anti-FMDV antibodies in the serum increase with increasing dose of vaccine administered.

Specific points:

1. Line 70 : The authors refer to "post-vaccination 146S content". Is this an error?

2. Section 2.4.2 is not explained very clearly and could be improved. Sample type should be specified (serum?)

3. Figure legends should be added to each figure.

4. Line 184 :  "146S contents" should be changed to "dose" 

5. The titles of Figures 1, S1, 3, and 5 should be changed from "varying 146S vaccine concentrations" to "varying vaccine doses". The authors do not say that the vaccines were diluted, so it is assumed that a smaller volume was given for the 1/3 and 1/9 doses. Therefore, the concentration is the same, it's just the amount that is administered (dose) that is varied.

6. The units on the x axes in Figures 2, 4, and 6 should be changed from ug/mL to ug.

7. Figure 3 - include A serotype in title.

8. The title of Figure 5 should be changed from "IFN-γ secretion levels" to "Serum IFN-γ levels" to make it clearer.

9. Lines 241-3: "Research has shown that measuring the level of IFN-γ secretion has the potential to replace the challenge test in assessing the effectiveness of FMD vaccines[31]" Reference 31 measures IFN gamma secretion from splenocytes taken from vaccinated mice that were restimulated with FMDV antigen in vitro. Therefore, this is a direct measure of IFN-gamma release by FMDV-specific T cells, not just a measurement of general serum IFN gamma. This reference does not claim that measurement of serum IFN gamma has the potential to replace the PD50 test for assaying vaccine effectiveness. This sentence should be re-worded so as not to imply that it does.

10. Lines 281-2 - insert a sentence to state that the PD50 test was performed.

11. It seems obvious that the 146S content would directly correlate with PD50, since both are a measure of the quantity of antigen present in the vaccine.

Comments on the Quality of English Language

Overall good. Some minor spelling and punctuation errors to be corrected during editing.

Round 2

Reviewer 3 Report

Comments and Suggestions for Authors

Thank you for taking the time to address my concerns about the manuscript. I feel the manuscript has been sufficiently improved to warrant publication in Veterinary Sciences.